# Patient Education on Exercise Prehabilitation Among Patients Receiving Neoadjuvant Therapy for Cancer Surgery in China: A Mixed-Methods Study

**DOI:** 10.3390/healthcare13050477

**Published:** 2025-02-22

**Authors:** Xiaohan Xu, Jiao Zhang, Yuelun Zhang, Tianxue Yang, Xuerong Yu

**Affiliations:** 1Department of Anesthesiology, Peking Union Medical College Hospital, Chinese Academy of Medical Science and Peking Union Medical College, Beijing 100730, China; xxu11@bidmc.harvard.edu (X.X.); zhangjiao722@pumch.cn (J.Z.); 2Department of Anesthesia, Critical Care and Pain Medicine, Center for Anesthesia Research Excellence (CARE), Beth Israel Deaconess Medical Center, Harvard Medical School, Boston, MA 02215, USA; 3Center for Prevention and Early Intervention, National Infrastructures for Translational Medicine, Institute of Clinical Medicine, Peking Union Medical College Hospital, Chinese Academy of Medical Science and Peking Union Medical College, Beijing 100730, China; zhangyuelun@pumch.cn; 4School of Sports Medicine and Physical Therapy, Beijing Sport University, Beijing 100084, China; 2024012970@bsu.edu.cn

**Keywords:** exercise prehabilitation, patient education, neoadjuvant therapy, mixed-methods, facilitator, barrier

## Abstract

**Background/Objectives**: Patients undergoing neoadjuvant therapy have ample time to engage in exercise prehabilitation. This study aimed to describe the current status, facilitators, and barriers of exercise prehabilitation among this population. **Methods**: This sequential explanatory mixed-methods evaluation was conducted at a general tertiary hospital in Beijing. It included a quantitative survey of patients who received neoadjuvant therapy before cancer surgery and qualitative semi-structured interviews with both patients and physicians. Thematic analysis was conducted using the Capability, Opportunity, and Motivation Behavior model. **Results**: A total of 269 patients participated in the survey, with a completion rate of 99.6%. Only 52.6% and 1.1% of patients met the standards for aerobic and muscle-strengthening activities, respectively. Fewer than 40% of patients reported learning about exercise prehabilitation from physicians. Patients’ knowledge was associated with meeting aerobic activity standards after adjusting for confounders (Level 1: odds ratio [OR] of 2.06, 95% confidence interval [CI] of 1.02–4.22; Level 2: OR of 2.56, 95% CI of 1.25–5.36). In total, 28 participants were interviewed. Facilitators of patient education on exercise prehabilitation included the surgeon’s ability to gain trust and patients’ prior commitment to exercise. Barriers included physicians’ lack of awareness of exercise benefits, insufficient knowledge or time for patient education, concerns about patients’ exercise ability, lack of referrals to rehabilitation clinics, challenges in follow-up, conflicts with cultural beliefs, and inadequate insurance coverage. **Conclusions**: This study revealed a lack of physician-led patient education on exercise prehabilitation. Efforts are needed to enhance physician education, implement collaborative clinics, and provide remote supervision.

## 1. Introduction

Multimodal prehabilitation includes exercise, nutritional supplementation, and psychological interventions, aiming to optimize functional capacity and mitigate the metabolic response to surgery [1]. Previous evidence suggested that multimodal prehabilitation is beneficial for reducing complications, enhancing recovery, and prolonging disease-free survival for colorectal, lung, orthopedic, and transplant surgeries [2,3,4,5,6,7]. Notably, patients who failed to adhere to the exercise plan were excluded in most of these trials. The potential efficacy of exercise prehabilitation includes improving functional capacity, optimizing organ perfusion, and promoting anti-tumor defense [8]. However, in real clinical practice, the effect of exercise prehabilitation may be less pronounced, as compliance with the proposed interventions can be as low as 16% [9,10]. A lack of education and guidance from healthcare providers can hinder patients from participating in prehabilitation [11].

Neoadjuvant therapy, including chemotherapy, radiation therapy, immunotherapy, and targeted therapy, has proven effective in enhancing tumor resectability and preserving organs during cancer surgery, leading to an improved rate of complete resection [12,13]. In recent years, neoadjuvant therapy has been increasingly utilized in the treatment of breast, esophageal, gastric, pancreatic, rectal, extremity sarcoma, bladder, and ovarian cancers [14]. Neoadjuvant therapy typically spans several weeks to months, providing patients with sufficient time to engage in prehabilitation [15]. Previous studies have illustrated facilitators and barriers of exercise prehabilitation among elderly or frail patients [16,17,18,19]. However, evidence is limited among patients undergoing neoadjuvant therapy. How effectively patients perform exercise prehabilitation during or after neoadjuvant therapy remains uncertain. Furthermore, patients receive care from physicians across various departments during and after neoadjuvant therapy, including surgeons, oncologists, and anesthesiologists. It is unclear how this multidisciplinary team collaborates on patient education for exercise prehabilitation and what challenges they may encounter.

The aim of this study was to describe the current status of exercise prehabilitation among patients receiving neoadjuvant therapy prior to cancer surgery and to identify the facilitators and barriers to patient education on exercise prehabilitation. Understanding these factors can help us identify areas for improvement and develop targeted strategies to enhance exercise prehabilitation.

## 2. Materials and Methods

This study was a single-center, sequential explanatory, mixed-methods evaluation. We first conducted a quantitative survey, followed by qualitative semi-structured interviews to explain the survey results. This study was approved by the institutional review board of Peking Union Medical College Hospital (No. K6176), and written informed consent was obtained from all participants in both the quantitative and qualitative phases. The quantitative component was reported in accordance with the Strengthening the Reporting of Observational Studies in Epidemiology (STROBE) checklist, while the qualitative component followed the Consolidated Criteria for Reporting Qualitative Research (COREQ) guidelines.

### 2.1. Study Context

This study was conducted at Peking Union Medical College Hospital, a general tertiary teaching hospital in Beijing, China, between May to September 2024. Patients diagnosed with cancer typically visited the surgeons’ outpatient clinics to determine whether their condition could be treated with surgery. After assessing the progression of their cancer, the surgeons sometimes recommended neoadjuvant therapy prior to surgery. Patients then visited medical and/or radiation oncologists, either at our hospital or at their local hospitals. Upon completing the neoadjuvant therapy, they returned to the surgeons’ outpatient clinics. After reassessing the patients, the surgeons would typically schedule the surgery two to three weeks later and, in some cases, refer patients to the anesthesiologist’s outpatient clinic for preoperative evaluation and optimization. Patients were typically admitted to the hospital wards one to three days before surgery. During this process, surgeons, oncologists, and anesthesiologists could recommend patients to rehabilitation physicians’ outpatient clinics if they deemed it necessary. Rehabilitation physicians evaluated cardiorespiratory function, developed individualized exercise plans, and provided exercise supervision for the patients.

### 2.2. Quantitative Study Population

The quantitative survey was distributed to all adult patients who received neoadjuvant therapy and were scheduled for cancer surgery between 22 May 2024 and 15 September 2024. The neoadjuvant therapy included chemotherapy, radiation therapy, immunotherapy, and targeted therapy, administrated either alone or in combination. The study population was identified by reviewing the medical charts where details of neoadjuvant therapy were recorded. Patients who refused to participate or did not complete the survey were excluded.

### 2.3. Quantitative Data Collection

Two anesthesiologists, a surgeon, an oncologist, and a rehabilitation physician jointly developed an open and anonymous questionnaire with 22 questions, including single-choice, multiple-choice, and 5-point Likert scale items, using an online survey tool (https://www.wjx.cn, accessed on 1 May 2024). Adaptive questioning was applied to certain items. These physicians discussed whether each item should be included in the questionnaire over three rounds until they reached a consensus. Two anesthesiologists conducted a pilot survey by asking each question to 15 patients, and the language was refined based on the patients’ feedback.

The questions aimed to collect data on aerobic activity (both moderate and vigorous intensity) and muscle-strengthening activity before, during, and after neoadjuvant therapy (Appendix A). Quality of life (QoL) during and after therapy was assessed using the EuroQol-5 Dimensions-3 Levels (EQ-5D-3L) instrument (Questions 10 and 17), while pain was measured using the 0–10 Numeric Rating Scale (NRS) (Questions 18) [20,21]. Patients’ perceived importance of exercise prehabilitation was compared to nutritional and psychological prehabilitation using a 5-point Likert scale (Question 19). Their knowledge of exercise prehabilitation was categorized into four levels: Level 0 indicated no awareness of its importance; Level 1 indicated awareness without specific knowledge; Level 2 indicated awareness with limited understanding of the type, volume, and intensity of exercise prehabilitation; and Level 3 indicated comprehensive knowledge (Question 20). We also asked about their sources of information, including physicians from various departments (Question 21), as well as potential barriers to exercise, such as physical discomfort, concerns about worsening their condition, inconvenient environments, lack of time, and negative emotions (Question 22).

In the full-scale survey, the same two anesthesiologists from the pilot survey asked and explained to the included patients in the holding area before surgery. Many patients could not distinguish between vigorous-intensity aerobic, moderate-intensity aerobic, and muscle-strengthening activities. As a result, we asked them to list all the exercises they performed, and we categorized them ourselves. Moderate-intensity aerobic activities included brisk walking or cycling. Examples of vigorous-intensity aerobic activities were jogging, mountain climbing, and carrying heavy groceries [22]. If we were unsure about the intensity, we used the talk test for categorization. Being able to speak in full sentences but not sing indicated moderate intensity, while difficulty speaking comfortably signified vigorous intensity [23]. Typical muscle-strengthening activities included lifting weights, using resistance bands, or performing body weight exercises [22]. In addition, we collected data on demographic information, cancer types, neoadjuvant therapy types, and comorbidities from electronic medical charts.

### 2.4. Quantitative Data Analysis

We first described the current practice of exercise prehabilitation, including the percentage of patients who engaged in aerobic and muscle-strengthening activities from the completion of neoadjuvant therapy to the day of surgery, as well as the percentage of patients whose exercise volume met the standards outlined in the “Physical Activity Guidelines for Americans” [22]. The standards were at least 150 min of moderate-intensity aerobic activity per week and muscle-strengthening activity at least 2 days per week [22]. One minute of vigorous-intensity aerobic activity was equivalent to two minutes of moderate-intensity aerobic activity [22]. We focused primarily on the period following the completion of neoadjuvant therapy, as patients’ exercise ability could have been limited by physical discomfort during treatment which generally alleviated after the completion of therapy. Patients’ perceived importance of exercise was compared to that of nutritional supplementation and psychological support using the Kruskal–Wallis test. Additionally, the sources of their knowledge and potential barriers to engaging in exercise prehabilitation were analyzed by the chi-square test.

In addition, we investigated the association between patients’ knowledge and their practice of exercise prehabilitation, with knowledge level as the exposure and meeting the standards for aerobic activity after neoadjuvant therapy as the outcome. Muscle-strengthening activity was not included in the outcome due to the small number of patients meeting the muscle-strengthening standard. Confounders that were potentially related to both the knowledge and practice, based on our experience, were selected, including age, sex, body mass index (BMI), pre-illness exercise habits that met the standards for aerobic activity, QoL during and after therapy (categorized using the median EQ-5D-3L score as the threshold), moderate or severe pain (NRS ≥ 4), hypertension, coronary artery disease, diabetes, and cancer type (digestive, gynecological, or others). The distribution of confounders was assessed using standardized mean difference (SMD), with an SMD > 0.1 considered indicative of imbalance. Two multivariable logistic regression models were used to adjust for confounding effects: one included all confounders, while the other included only the unbalanced confounders.

Statistical analysis was conducted using R (R version 4.2.1; R Foundation for Statistical Computing, Vienna, Austria, 2018), along with tableone and ggplot2 packages. A two-sided *p* < 0.05 indicated statistical significance.

### 2.5. Qualitative Study Population

We employed a nonprobabilistic, maximum variation sampling strategy to recruit interview participants, ensuring representation from all parties involved in neoadjuvant therapy. Patients who had completed neoadjuvant therapy, along with their family members, were included. Additionally, we interviewed healthcare professionals from various departments who cared for neoadjuvant therapy, including surgeons, oncologists, anesthesiologists, rehabilitation physicians, physical therapists, and nurses. To achieve maximum variation, we selected patients either actively or inactively engaging in exercise prehabilitation, surgeons from different cancer subspecialties, oncologists from both our hospital and a secondary hospital within the same network as our hospital, and anesthesiologists with or without a research focus on prehabilitation. The physicians included in this study were also diverse in age, sex, academic titles, and work experience. Recruitment ceased when additional interviews no longer generated new codes or themes, indicating data saturation [24].

### 2.6. Qualitative Data Collection

We inductively developed three interview guides, each comprising 9 to 11 open-ended questions, tailored for patients/family members, surgeons/oncologists/anesthesiologists, and rehabilitation physicians/physical therapists/nurses, respectively (Appendix A). The interview questions focused on the facilitators and barriers to implementing exercise prehabilitation for patients, as well as the challenges and strategies healthcare professionals faced in conducting patient education on exercise prehabilitation. The interview guides were piloted with a patient, a surgeon, and an anesthesiologist, and subsequent revisions were made based on their feedback. Concurrent with data analysis, ongoing discussions were held after each interview to refine the questions.

Semi-structured interviews were conducted face-to-face with each participant at their convenience, either in outpatient clinics or wards, by two researchers: Y.X., MD (a female anesthesiologist), and Z.Y., PhD (a male epidemiologist experienced in qualitative research), both of whom worked at Peking Union Medical College Hospital. Regardless of the location, no additional individuals were present during the interviews. At the outset, participants were briefed on this study’s purpose. All interviews were audio-recorded with written informed consent obtained, and the transcriptions were conducted verbatim, with participant names replaced by assigned numbers to maintain anonymity.

### 2.7. Qualitative Data Analysis

A thematic analysis of the interview transcripts was conducted using the Capability, Opportunity, and Motivation Behavior (COM-B) model from the Behavior Change Wheel [25]. The COM-B model has been recognized as a useful and systemic framework for identifying determinants of behavior and facilitators or barriers to behavior change and has been widely used in previous studies on similar research questions [25,26]. In the context of our study, capability referred to factors that enabled or hindered exercise prehabilitation, motivation encompassed factors influencing willingness or perceived importance of exercise prehabilitation, opportunity included policy or environmental factors affecting its implementation, and behavior involved approaches to promoting exercise prehabilitation.

Two investigators (X.X. and Y.T.), who were not involved in the interview process, employed a deductive, line-by-line approach to generate initial codes and searched for themes by identifying connections between codes [24]. Weekly meetings were held, during which the other two investigators (Y.X. and Z.Y.) reviewed the codes and themes developed by X.X. and Y.T., assessed their interrater reliability, discussed any discrepancies, conducted data source triangulation by comparing codes from various participants, and ultimately mapped the themes onto the COM-B model [24,26]. The model was continuously refined throughout the study as new data emerged from concurrent interviews. Notably, X.X. and Y.X., both anesthesiologists, had professional and personal affiliations with some participants (insiders). To mitigate potential insider bias, reflexivity memo writing was employed, allowing researchers to continuously document their thoughts, biases, and positionality throughout the research process [27]. This practice helped maintain self-awareness and ensured that interpretations were data-driven rather than influenced by prior relationships or assumptions. Additionally, peer debriefing with Y.T. and Z.Y., who were not anesthesiologists (outsiders), provided an external perspective, challenging assumptions and reducing the risk of misinterpretation due to pre-existing researcher biases. Five participants, including a patient, a surgeon, an oncologist, an anesthesiologist, and a rehabilitation physician, were invited to review the thematic model, and minor revisions were made based on their feedback. The qualitative data analysis utilized NVivo, version 12 (QSR International, Melbourne, Australia).

## 3. Results

### 3.1. Quantitative Results

A total of 269 patients received neoadjuvant therapy and underwent cancer surgery between May and September 2024 at Peking Union Medical College Hospital. One patient was excluded for not completing the survey on exercise prehabilitation, resulting in a 99.6% completion rate and leaving 268 patients [mean age (standard deviation, SD): 58 (11) years, 38.3% males] included in the quantitative analysis (Figure 1). Among them, 101 patients (38.7%) underwent gynecological surgery, 80 patients (30.7%) underwent colorectal surgery, and 26 patients (10.0%) underwent gastric surgery. A total of 247 patients (94.6%) received chemotherapy, 57 patients (21.8%) underwent radiation therapy, 47 patients (18.0%) received immunotherapy, and 36 patients (13.8%) had targeted therapy.

Of the patients included, 170 patients (63.4%) engaged in aerobic activity from the completion of neoadjuvant therapy until the day of surgery, with 141 patients (52.6%) meeting the “Physical Activity Guidelines for Americans” (Figure 1). All of these 170 patients performed moderate-intensity activity, and 8 additionally engaged in vigorous-intensity aerobic activity. In contrast, only four patients (1.5%) participated in muscle-strengthening activities during this period, with three patients (1.1%) meeting the recommended standards (Figure 1). The Kruskal–Wallis test indicated that the perceived importance of exercise [median (interquartile range, IQR): 3 (1–4)] was significantly lower than that of nutritional supplementation or psychological support [both median (IQR): 4 (3,4)] (*p* < 0.001, Figure 2A). The largest number of patients (120, 44.8%) identified friends or family as their source of knowledge on exercise prehabilitation, followed by oncologists (103, 38.4%) and surgeons (95, 35.4%) (Figure 2B). Notably, only five patients (1.9%) reported learning from rehabilitation physicians. The three leading barriers to implementing exercise prehabilitation were physical discomfort (231 patients, 86.2%), concern about worsening conditions (73 patients, 27.2%), and negative emotions (37 patients, 13.8%) (Figure 2C).

In total, 130 patients (48.5%) were unaware of the importance of exercise prehabilitation (level 0), 71 patients (26.5%) recognized its importance but lacked knowledge on its implementation (level 1), 64 patients (23.9%) had limited knowledge of how to implement it (level 2), and only 3 patients (1.1%) had comprehensive knowledge (level 3). Level 3 and level 2 were combined in the subsequent association analysis due to the limited number of patients in level 3. Most potential confounders were unbalanced across the different knowledge levels, except for age, sex, and hypertension (Table 1). Knowledge of exercise prehabilitation was significantly associated with meeting the standards for aerobic activity in the logistic regression models, whether adjusting for no confounders [level 1: odds ratio (OR) of 1.62, 95% confidence interval (CI) of 0.67–3.90, *p* = 0.280; level 2–3: OR of 2.21, 95% CI of 1.30–3.82, *p* = 0.004], all confounders (level 1: OR of 2.06, 95% CI of 1.02–4.22, *p* = 0.044; level 2–3: OR of 2.56, 95% CI of 1.25–5.36, *p* = 0.011), or only unbalanced confounders (level 1: OR of 2.13, 95% CI of 1.06–4.34, *p* = 0.035; level 2–3: OR of 2.59, 95% CI of 1.27–5.38, *p* = 0.010) (Table 2). These findings indicate that patients’ likelihood of meeting the exercise standards improved as their knowledge increased.

### 3.2. Qualitative Results

A total of twenty-eight participants were interviewed, including four patients, one patient’s family members, fifteen surgeons specializing in thoracic, gynecological, general, urological, breast, and hepatic surgeries, two oncologists, three anesthesiologists, one rehabilitation physician, one physical therapist, and one nurse. Their sex, age, and other characteristics were provided in Appendix A. The interview lasted between 17 and 42 min. Overall, 15 subthemes were identified and mapped to the COM-B model (Figure 3). Appendix A provided quotations for each subtheme, demonstrating consistency in the data across participants with different roles.

#### 3.2.1. Capacity

Among all the healthcare providers, surgeons were the most trusted by patients: “My surgeon is the person I trust the most. If he told me: ‘You must exercise to a certain level, or we won’t proceed with the surgery’, I definitely wouldn’t be this lazy” (participant 3, patient). Education provided by surgeons could effectively improve patients’ adherence: “Later, the thoracic surgeons began educating patients about the importance of prehabilitation, and things became earlier” (participant 26, rehabilitation physician).

Despite patients’ trust, few surgeons, oncologists, and anesthesiologists offered education on exercise prehabilitation possibly due to a lack of knowledge, time, or energy: “I am not very familiar with some specialized exercises, so I’m unable to provide specific instructions, such as pelvic floor exercises for colorectal or gynecological surgery” (participant 23, anesthesiologist); “We focus on performing surgeries well and minimizing the risk of complications. We don’t have time to address exercise and nutrition in detail” (participant 12, surgeon).

Consistent with the quantitative data, patients who were committed to regular exercise before their illness showed a stronger desire and ability to continue exercising during or after neoadjuvant therapy: “I have been doing farm work since I was young. I can’t stay idle. Even after getting sick, walking for one hour a day is not a difficult task for me” (participant 1, patient). Therefore, greater attention and education from physicians was needed for patients without an established exercise habit.

#### 3.2.2. Motivation

Nearly all surgeon participants viewed exercise as less important than nutrition and mental health, suggesting a lack of awareness about its benefits. These findings aligned with patients’ perceptions of the importance of prehabilitation components in the quantitative analysis, highlighting the significant influence physicians had on their patients’ views.

For patients in poor condition, some physicians expressed concerns about their ability to tolerate exercise: “Patients who come to me are usually quite weak, often suffering from anemia and poor nutrition, making exercise challenging” (participant 13, surgeon). However, they did not recognize that these patients could recover through nutritional supplementation or after completing chemotherapy. Conversely, for patients in seemingly fit condition, physicians often overlooked their potential lack of physical fitness: “We often believe our patients typically do not have significant issues with fitness; however, reduced physical strength from chemotherapy are generally not visible” (participant 16, surgeon).

#### 3.2.3. Opportunity

Physicians recognized the need to seek assistance from professionals for prehabilitation, given their limited knowledge and time, as previously discussed. However, they rarely referred patients to rehabilitation outpatient clinics due to the absence of a standardized referral process. In contrast, it was more convenient to order a rehabilitation consultation for hospitalized patients: “Most patients have the rehabilitation department involved after being admitted to the hospital for surgery. It’s rare for them to get involved in outpatient clinics during neoadjuvant therapy. Unfortunately, the preoperative hospital stay is relatively short, so there’s only 2–4 days for practice.” (participant 27, physical therapist).

Another major barrier to exercise prehabilitation is the lack of follow-up and supervision for patients before surgery, as some lived far from Beijing and received neoadjuvant therapy at their local hospitals: “Many patients are from other provinces or rural areas and are unwilling to come all the way to Beijing just for exercise rehabilitation guidance. Even if they visit the rehabilitation outpatient clinic, it’s usually on the same day as their surgical or chemotherapy appointments, meaning they often receive in-person guidance only once—and one session is often not enough” (participant 26, rehabilitation physician).

Exercise prehabilitation was particularly challenging in China, partly due to traditional cultural beliefs that promote complete rest during illness: “After getting sick, I still want to do exercise, but I hesitate because people around you constantly warn: ‘Don’t push yourself too hard—what if something goes wrong?’” (participant 2, patient). Furthermore, prehabilitation-related costs were not covered by medical insurance in China: “Patients may feel that out-of-pocket expenses are unnecessary and may doubt the doctor’s intentions” (participant 7, surgeon).

#### 3.2.4. Behavior

A few physicians acknowledged their lack of awareness regarding the basic concepts of exercise prehabilitation and emphasized the need for physician education: “It’s more effective to educate doctors first than to educate patients” (participant 21, oncologist). Furthermore, prehabilitation relies on a multidisciplinary team, and a combined outpatient clinic could effectively address the challenges associated with patient referrals: “The goal is to streamline care and prevent patients from having to visit multiple clinics separately” (participant 11, surgeon).

To address the challenges of follow-up during neoadjuvant therapy, participants suggested implementing remote exercise supervision via online platform: “We are experimenting with remote guidance, including question and answer, a video library of exercise demonstrations, and check-ins” (participant 26, rehabilitation physician). Additionally, a simplified plan with measurable goals could improve patient adherence and reduce the need for physician supervision: “We ask patients to wear a fitness watch and keep their heart rate within a specified range” (participant 6, surgeon).

## 4. Discussion

This mixed-methods study revealed a low percentage of patients meeting the exercise standards and an inadequate understanding of the importance of exercise prehabilitation among patients receiving neoadjuvant therapy. Facilitators of patient education on exercise prehabilitation included the surgeon’s ability to gain patient trust and patients’ prior commitment to exercise. Barriers included physicians’ lack of awareness of exercise benefits, insufficient knowledge or time for patient education, concerns about patients’ exercise ability, lack of standardized referrals to rehabilitation clinics, challenges in follow-up and supervision, conflicts with cultural beliefs, and inadequate insurance coverage.

Our findings revealed a significant gap in patient education on exercise prehabilitation provided by physicians. Fewer than 40% of patients reported receiving information on exercise prehabilitation from their physicians, a percentage even lower than that from family or friends. Knowledge levels were associated with the likelihood of meeting prehabilitation standards, suggesting that inadequate education from physicians may contribute to suboptimal exercise prehabilitation practices, consistent with findings from previous studies [16,17,18,19].

Patient education on exercise prehabilitation relies on a multidisciplinary team approach [28]. Although rehabilitation physicians and therapists were the most qualified to offer specific guidance on exercise prehabilitation, patients often did not realize they needed to visit the rehabilitation department unless recommended or referred by their surgeons, oncologists, or anesthesiologists. However, a lack of knowledge, time, and energy presented significant challenges to these physicians. Our results showed that surgeons were the most trusted physicians by patients during neoadjuvant therapy, and their educational efforts were the most effective. Unfortunately, however, many of the surgeons in our study missed valuable opportunities to emphasize the importance of exercise prehabilitation, possibly due to their own lack of understanding of its benefits. This underscores the need for improved physician education. Another reason was that surgeons, oncologists, and anesthesiologists had limited time for each patient in their outpatient clinics, which prevented them from providing education on exercise. A recent randomized controlled trial demonstrated the effectiveness of group-based patient education, presenting another potential solution to addressing the limitation of physicians’ time and energy for individualized patient [29]. Furthermore, effort should be made to simply the referral process or implement collaborative clinics, making multidisciplinary collaboration less time-consuming and more continent, as suggested by our participants and a previous study [28].

Our data provided new insights into the challenges of following up with patients during the intercycle periods of neoadjuvant therapy. Physical symptoms and comorbidities were widely recognized as barriers to prehabilitation [11,19,29]. Patients might experience adverse effects and discomfort during chemotherapy, which can present a significant challenge to engaging in exercise, as reported in our survey and a prior study [30]. Additionally, physicians expressed concerns about patients’ tolerance and the safety of exercise during the interviews. It is worth noting that discomfort could ease or alleviate during the intercycle period and after the completion of therapy. Unfortunately, patients often did not visit clinics during this time, making it challenging to follow up with them and fully utilize this period for prehabilitation. In line with previous evidence, our participants suggested implementing remote exercise instruction and supervision through online platforms [11,31,32,33,34]. The online platforms should offer personalized plans adoptable to local settings, video demonstrations, adherence tracking, and direct communication with physicians. Support from family members was also highly valued in this process, highlighting the need to educate them as well [18]. Additional attention and protection should be given to patients without prior exercise habits, as they may face greater challenges in implementing exercise plans, particularly muscle-strengthening activities. They also face a higher risk of injury if they struggle to understand remote exercise supervision.

Our study had several strengths. First, to our knowledge, it was the first mixed-methods study focusing on patient education in exercise prehabilitation. The quantitative and qualitative data from participants in different roles demonstrated consistency at multiple points, indicating the robustness of the findings. Second, we explained each survey question to patients face-to-face, achieving a high completion rate and minimizing the risk of information bias. Finally, we recruited physicians from various departments and patients for the qualitative study, allowing us to gather objective and compelling evidence. These strengths enabled our study to provide a comprehensive understanding of the facilitators, barriers, and potential solutions—some of which have seldom been identified in previous research.

Our findings should be interpreted with caution due to several limitations. First, our conclusions may not be applicable to centers in different contexts, as data were collected from a single general referral medical center in China. Multi-center studies with larger sample sizes across diverse cultural backgrounds are warranted in the future. Second, the limited sample size of the survey could hinder representativeness, despite our efforts to include patients from various departments. Third, the standards outlined in the “Physical Activity Guidelines for Americans” were used in this study to define whether participants met the exercise standards. However, these guidelines were not originally developed for prehabilitation among neoadjuvant therapy patients and may need to be adapted based on their cardiorespiratory tolerability and the fitness requirements of major surgeries. Including respiratory muscle strength and stretching exercises is also warranted. Third, the exercise performance data were collected through survey, which might introduce information bias. Objective measurement using wearable devices is recommended for future studies. Finally, patient education on exercise prehabilitation involves not only hospitals but also rehabilitation institutes and community settings. Future studies are warranted to provide insights from perspectives beyond hospitals.

## 5. Conclusions

In conclusion, this study illustrated that the current practice of patient education on exercise prehabilitation requires significant improvement. To address the identified barriers, key stakeholders are encouraged to prioritize initiatives such as enhancing physician education, implementing collaborative clinics, and providing remote supervision.

## Figures and Tables

**Figure 1 healthcare-13-00477-f001:**
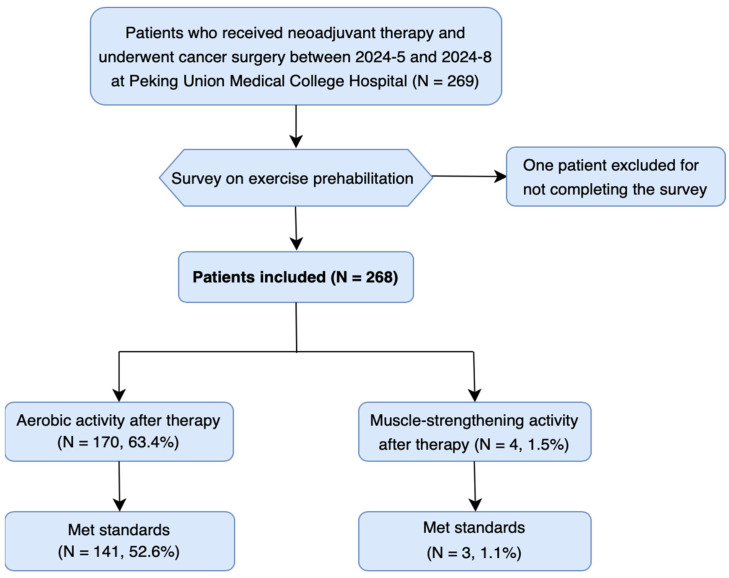
Flow diagram of the selection of the patients included in the qualitative study and primary results.

**Figure 2 healthcare-13-00477-f002:**
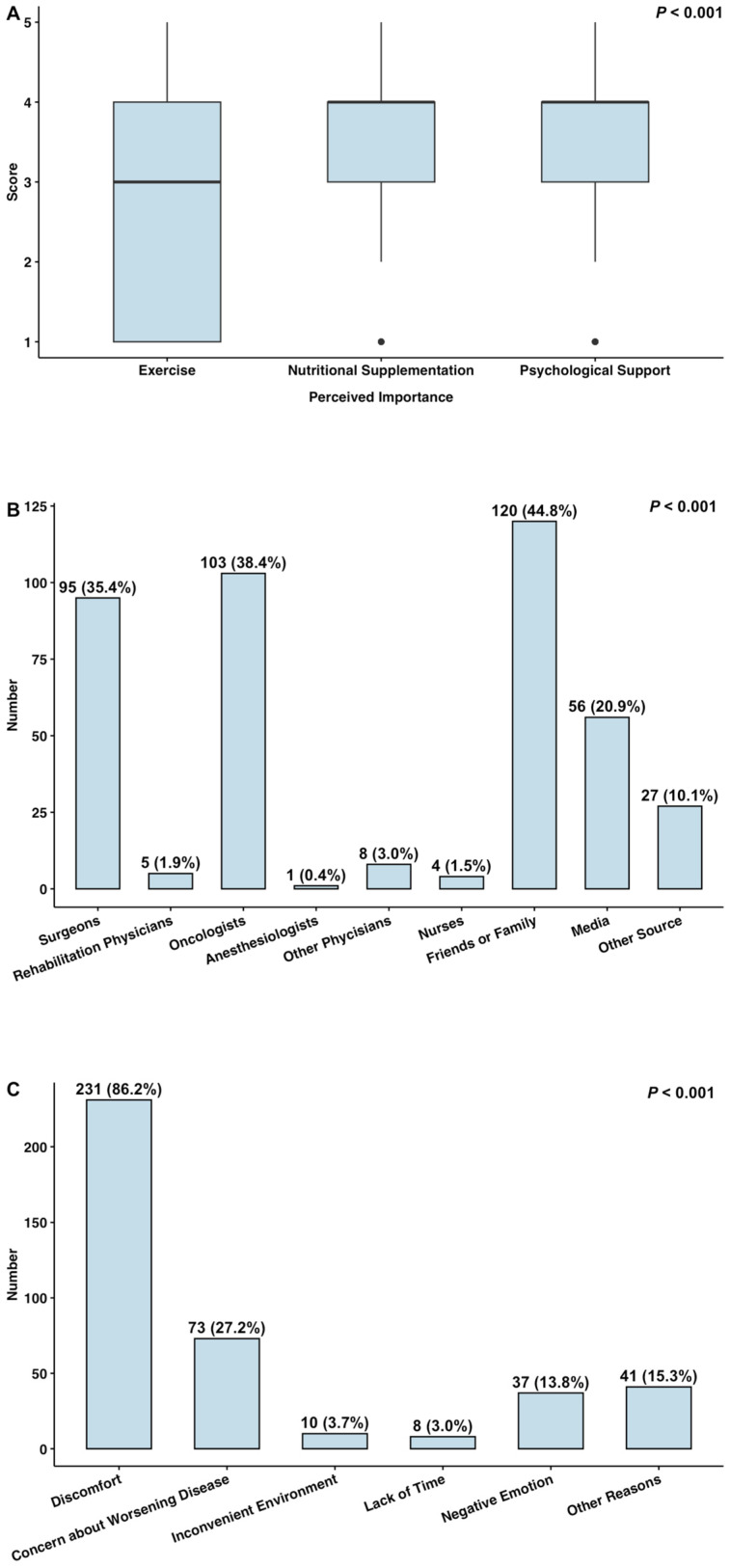
Results of the survey: (**A**) box plot of the perceived importance of exercise, nutritional supplementation, and psychological support; (**B**) bar plot of the source of knowledge on exercise prehabilitation; (**C**) bar plot of the factors influencing the implementation of exercise prehabilitation.

**Figure 3 healthcare-13-00477-f003:**
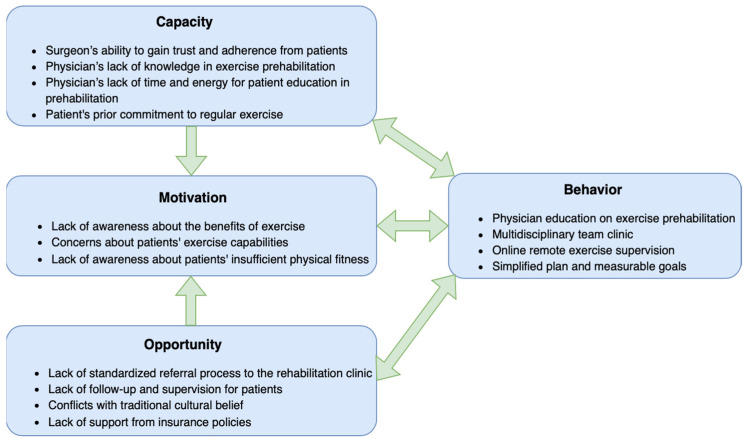
Capacity, Opportunity, Motivation Behavior model of the facilitators and barriers to exercise prehabilitation.

**Table 1 healthcare-13-00477-t001:** Distribution of confounders.

Variables	All Patients (N = 268)	Knowledge of Exercise Prehabilitation ^a^			SMD
		Level 0 (N = 130)	Level 1 (N = 71)	Level 2–3 (N = 67)	
Age (year)	58 (11)	58 (12)	57 (11)	58 (10)	0.015
Sex (male)	100 (38.3%)	47 (37.6%)	28 (39.4%)	25 (38.5%)	0.025
BMI (kg/m^2^)	23.3 (4.5)	24.0 (4.9)	22.4 (5.0)	23.1 (2.8)	0.247
Previous exercise habit	126 (47.0%)	49 (37.7%)	38 (53.5%)	39 (58.2%)	0.279
QoL during therapy (EQ-5D-3L ≥ median)	139 (51.9%)	72 (55.4%)	35 (49.3%)	32 (47.8%)	0.102
QoL after therapy (EQ-5D-3L ≥ median)	98 (36.6%)	48 (36.9%)	34 (47.9%)	16 (23.9%)	0.342
Moderate or severe pain	30 (11.2%)	17 (13.1%)	9 (12.7%)	4 (6.0%)	0.163
Hypertension	60 (23.0%)	28 (22.4%)	16 (22.5%)	16 (24.6%)	0.035
Coronary artery disease	9 (3.4%)	5 (4.0%)	1 (1.4%)	3 (4.6%)	0.126
Diabetes	33 (12.6%)	19 (15.2%)	8 (11.3%)	6 (9.2%)	0.122
Cancer type					0.304
Digestive cancer	128 (49.0%)	57 (45.6%)	39 (54.9%)	32 (49.2%)	
Gynecological cancer	101 (38.7%)	54 (43.2%)	19 (26.8%)	28 (43.1%)	
Other cancer	32 (12.3%)	14 (11.2%)	13 (18.3%)	5 (7.7%)	

Normally and non-normally distributed continuous variables were described as mean (standard deviation) and categorical variables were reported as number (percentage). Abbreviations: SMD, standardized mean difference; BMI, body mass index; QoL, quality of life; EQ-5D-3L, EuroQol-5 Dimensions-3 Levels. ^a^ Level 0: Lack of awareness regarding the importance of prehabilitation; Level 1: Awareness of its importance, but without knowledge of the specifics; Level 2–3: Awareness with limited or comprehensive knowledge of the type, volume, and intensity of exercise prehabilitation.

**Table 2 healthcare-13-00477-t002:** Association between knowledge of exercise prehabilitation and the practice of aerobic activity after neoadjuvant therapy (N = 268).

Variable	Model 1 ^a^			Model 2 ^b^		
	OR	95% CI	*p*	OR	95% CI	*p*
Knowledge of exercise prehabilitation ^c^						
Level 0	1.00 (ref)	NA	NA	1.00 (ref)	NA	NA
Level 1	2.06	1.02–4.22	0.044	2.13	1.06–4.34	0.035
Level 2–3	2.56	1.25–5.36	0.011	2.59	1.27–5.38	0.010
Age (year)	0.99	0.96–1.02	0.445	NA	NA	NA
Sex (male)	0.76	0.33–1.70	0.507	NA	NA	NA
BMI (kg/m^2^)	1.02	0.95–1.09	0.590	1.02	0.94–1.09	0.660
Previous exercise habit	8.50	4.38–17.30	<0.001	7.62	4.15–14.59	<0.001
QoL during therapy						
EQ-5D-3L < median	1.00 (ref)	NA	NA	1.00 (ref)	NA	NA
EQ-5D-3L ≥ median	1.21	0.64–2.31	0.568	1.22	0.65–2.34	0.534
QoL after therapy						
EQ-5D-3L < median	1.00 (ref)	NA	NA	1.00 (ref)	NA	NA
EQ-5D-3L ≥ median	0.44	0.21–0.89	0.024	0.42	0.21–0.85	0.017
Moderate or severe pain	0.42	0.14–1.20	0.116	0.42	0.14–1.18	0.107
Hypertension	1.06	0.51–2.23	0.875	NA	NA	NA
Coronary artery disease	0.36	0.07–1.84	0.214	0.32	0.06–1.63	0.165
Diabetes	0.71	0.29–1.77	0.463	0.69	0.28–1.68	0.411
Cancer type						
Digestive cancer	1.00 (ref)	NA	NA	1.00 (ref)	NA	NA
Gynecological cancer	0.49	0.21–1.12	0.094	0.60	0.32–1.13	0.116
Other cancer	0.89	0.34–2.34	0.803	1.00	0.40–2.52	0.998

Abbreviations: OR, odds ratio; CI, confidence interval; ref, reference; NA, not applicable; BMI, body mass index; QoL, quality of life; EQ-5D-3L, EuroQol-5 Dimensions-3 Levels. ^a^ Model 1 included all confounders. ^b^ Model 2 included only unbalanced confounders. ^c^ Level 0: Lack of awareness regarding the importance of prehabilitation; Level 1: Awareness of its importance, but without knowledge of the specifics; Level 2–3: Awareness with limited or comprehensive knowledge of the type, volume, and intensity of exercise prehabilitation.

## Data Availability

The data presented in this study are available upon request from the corresponding author due to the sensitive nature of the data collected for this study.

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
