# Peer review of "Patient Education on Exercise Prehabilitation Among Patients Receiving Neoadjuvant Therapy for Cancer Surgery in China: A Mixed-Methods Study"

_healthcare, 2025, doi:10.3390/healthcare13050477_

Round 1

Reviewer 1 Report

Comments and Suggestions for Authors

Thank you for providing me the opportunity of reviewing manuscript entitle” Patient Education on Exercise Prehabilitation among Patients Receiving Neoadjuvant Therapy for Cancer Surgery in China: a Mixed-Methods Study. The title is interesting, however some points are raised.

Abstract

Line 39: It is recommended that conclusion be revised according to results.

 Introduction

- Line 55: The transition from discussing multimodal prehabilitation to focusing specifically on exercise prehabilitation may benefit from a clearer delineation of how exercise fits within the broader context of prehabilitation.

-Line 61:  The discussion on neoadjuvant therapy is informative, but consider providing specific examples of the types of cancers being treated for more clarifications.

Methods

Study Design and Approval:

- Line 71: The description of the study as a "single-center, sequential explanatory, mixed-methods evaluation" is clear. However, consider briefly explaining what "sequential explanatory" entails for readers who may not be familiar with the terminology.

- Line 104: The development of the questionnaire with input from multiple specialists is very well, ensuring content validity. However, consider specifying how the items were chosen or developed before validation.

- Line 128: The methods for categorizing exercise types based on patient responses are necessary to elaborate on how this categorization was done to ensure consistency.

. Quantitative Data Analysis:

- Line 132: The analysis plan is thorough, detailing the statistical methods used. However, it would be helpful to clarify why certain confounders were selected, particularly those that are less common (e.g., specific cancer types, hypertension).

-- Clarifying the rationale for choosing specific interview settings (outpatient clinics or wards) could increase depth to the methodology.

- Line 196: the explanation of the COM-B model for thematic analysis is recommended. Consider providing a brief rationale for why this model was chosen over others.

- Knowledge Levels:

  - In the categorization of knowledge levels include the rationale for combining levels 2 and 3 in the association analysis could be more explicit.

- Logistic Regression Findings:

  - In the logistic regression models, consider adding a brief interpretation of the odds ratios to help readers understand their practical significance.

 Qualitative Results:

- Subthemes and COM-B Model:

  - Consider briefly explaining why the COM-B model was chosen for this analysis, as it provides a useful framework for understanding behavior change.

Discussion

Key Findings and Implications:

- The discussion of the gaps in physician education is well-articulated. It may strengthen the argument to include specific recommendations for educational interventions or training programs that could enhance physician knowledge about exercise prehabilitation.

Barriers to Implementation:

- It would be helpful to provide examples or evidence from the literature to support these claims, which would add depth to the discussion.

Strengths of the Study:

- The strengths outlined are compelling, particularly the mixed-methods approach and the high completion rate of surveys. It may be beneficial to briefly discuss how these strengths enhance the validity of the findings.

Limitations:

- It might enhance this section to suggest specific contexts or populations where future research could be conducted to validate or expand upon your findings.

- The comment about the use of the “Physical Activity Guidelines for Americans” is important. Consider providing a brief rationale for why these guidelines were chosen, despite their limitations, and suggest how future studies might develop tailored guidelines for this population.

Conclusions:

- It might be beneficial to include a call to action, urging stakeholders (e.g., healthcare providers, policymakers) to prioritize these changes as improvement in patient education regarding exercise prehabilitation..

Author Response

Dear Reviewer 1,

Please find the revised version of our manuscript (No.: 3463205) entitled “Patient Education on Exercise Prehabilitation among Patients Receiving Neoadjuvant Therapy for Cancer Surgery in China: a Mixed-Methods Study”. Thank you for offering us the opportunity to revise our manuscript for further consideration by the Healthcare.

We have revised the manuscript in response to the editors’ and reviewers’ comments. Our detailed responses to each of the comments are shown below. We would like to thank the editors and reviewers for their insightful comments, which have helped us improve the manuscript. We hope that this revised version addresses all comments adequately and that our manuscript is now considered acceptable for publication in Healthcare.

Sincerely,

Xuerong Yu, MD

-----------------------

Point-by-point Response to Comments of Reviewers

Reviewer 1

Comment:. Abstract: Line 39: It is recommended that conclusion be revised according to results.

Reply: As you suggested, we revised the conclusions as “The study revealed a lack of physician-led patient education on exercise prehabilitation. Efforts are needed to enhance physician education, implement collaborative clinics, and provide remote supervision” (line 37-40).

Comment: Introduction: Line 55: The transition from discussing multimodal prehabilitation to focusing specifically on exercise prehabilitation may benefit from a clearer delineation of how exercise fits within the broader context of prehabilitation.

Reply: Thank you for your suggestion. We added this in line 50-52 as follows: “The potential efficacy of exercise prehabilitation includes improving functional capacity, optimizing organ perfusion, and promoting anti-tumor defense.[8]”

Comment:  Introduction: Line 61: The discussion on neoadjuvant therapy is informative, but consider providing specific examples of the types of cancers being treated for more clarifications.

Reply: We agree with you, and provided examples of cancer types as “In recent years, neoadjuvant therapy has been increasingly utilized in the treatment of breast, esophageal, gastric, pancreatic, rectal, extremity sarcoma, bladder, and ovarian cancers [14]”. (line 59-61)

Comment: Methods: Study Design and Approval:- Line 71: The description of the study as a "single-center, sequential explanatory, mixed-methods evaluation" is clear. However, consider briefly explaining what "sequential explanatory" entails for readers who may not be familiar with the terminology.

Reply: Thank you for your constructive comments. “Sequential” means the qualitative analysis was conducted after quantitative analysis, and “explanatory” means the aim of qualitative analysis was to explain the results of quantitative analysis. We clarify this as “We first conducted a quantitative survey, followed by qualitative semi-structured interviews to explain the survey results” (line 77-78).

Comment: Methods: Line 104: The development of the questionnaire with input from multiple specialists is very well, ensuring content validity. However, consider specifying how the items were chosen or developed before validation.

Reply: We added this in line 113-115 as “These physicians discussed whether each item should be included in the questionnaire over three rounds until they reached a consensus”.

Comment: Methods: - Line 128: The methods for categorizing exercise types based on patient responses are necessary to elaborate on how this categorization was done to ensure consistency.

Reply: We appreciate your suggestion, clarified this in line 137-143 as “Moderate-intensity aerobic activities included brisk walking or cycling. Examples of vigorous-intensity aerobic activities were jogging, mountain climbing, and carrying heavy groceries. [22] If we were unsure about the intensity, we used the talk test for categorization. Being able to speak in full sentences but not sing indicated moderate intensity, while difficulty speaking comfortably signified vigorous intensity.[23] Typical muscle-strengthening activities included lifting weights, using resistance bands, or performing body weight exercises.[22]”.

Comment: Methods:. Quantitative Data Analysis:- Line 132: The analysis plan is thorough, detailing the statistical methods used. However, it would be helpful to clarify why certain confounders were selected, particularly those that are less common (e.g., specific cancer types, hypertension).

Reply: In this analysis, confounders should be related to both knowledge (exposure) and practice (outcome) but should not serve as intermediate variables between them. Confounders were typically selected based on researchers’ clinical experience and previous similar studies. Given the limited prior evidence on this research question, our selection was primarily guided by our experience. (line 165-166) Cancer type was included as a confounder because chemotherapy intensity varies across different cancers and may influence both the ability and willingness to acquire exercise knowledge and engage in physical activity. Additionally, hypotension and diabetes were considered, as physicians may provide specific recommendations for patients with these comorbidities to help them manage their conditions.

Comment:. Methods:. -- Clarifying the rationale for choosing specific interview settings (outpatient clinics or wards) could increase depth to the methodology.

Reply: The interviews were conducted at the participants' convenience (lines 202–203), either in outpatient clinics—typically after they had seen their last patient of the day—or in wards, usually in their offices. Regardless of the location, no additional individuals were present during the interviews (lines 205–206).

Comment: Methods:- Line 196: the explanation of the COM-B model for thematic analysis is recommended. Consider providing a brief rationale for why this model was chosen over others.

Reply: We explained this n line 213-215 as “The COM-B model has been recognized as a useful and systemic framework for identifying determinants of behavior and facilitators or barriers to behavior change, and has been widely used in previous studies on similar research questions.[25-26]”

Comment: Results:- Knowledge Levels: - In the categorization of knowledge levels include the rationale for combining levels 2 and 3 in the association analysis could be more explicit.

Reply: We explain this in line 268-269 as “Level 3 and level 2 were combined in the subsequent association analysis due to the limited number of patients in level 3”.

Comment: Results:- Logistic Regression Findings:- In the logistic regression models, consider adding a brief interpretation of the odds ratios to help readers understand their practical significance.

Reply: Thank you very much. We added this in line 278-279 as “These findings indicate that patients’ likelihood of meeting the exercise standards improved as their knowledge increased.”

Comment: Results: Qualitative Results:- Subthemes and COM-B Model:- Consider briefly explaining why the COM-B model was chosen for this analysis, as it provides a useful framework for understanding behavior change.

Reply: We agree that this rational is important, thus we explain this in Methods as “The COM-B model has been recognized as a useful and systemic framework for identifying determinants of behavior and facilitators or barriers to behavior change, and has been widely used in previous studies on similar research questions.[25-26] (line 213-215)

Comment: Discussion: Key Findings and Implications: - The discussion of the gaps in physician education is well-articulated. It may strengthen the argument to include specific recommendations for educational interventions or training programs that could enhance physician knowledge about exercise prehabilitation.

Reply: Thank you for your valuable suggestion. We conducted a thorough literature review but found no solid evidence supporting a physician training program on exercise prehabilitation. However, we identified a rigorous randomized controlled trial demonstrating the effectiveness of group-based patient education, which offers a potential solution to overcoming the barrier of limited physician time and energy for individualized patient education. We explain this in line 421-424 as “A recent randomized controlled trial demonstrated the effectiveness of group-based patient education, presenting another potential solution to addressing the limitation of physicians' time and energy for individualized patient.[29]”

Comment: Discussion: Barriers to Implementation:- It would be helpful to provide examples or evidence from the literature to support these claims, which would add depth to the discussion.

Reply: We cited reference 30-34 to supported our claims as “Patients might experience adverse effects and discomfort during chemotherapy, which can present a significant challenge to engaging in exercise, as reported in our survey and a prior study[30]. Additionally, physicians expressed concerns about patients’ tolerance and the safety of exercise during the interviews. It is worth noting that discomfort could ease or alleviate during the intercycle period and after the completion of therapy. Unfortunately, patients often did not visit clinics during this time, making it challenging to follow up with them and fully utilize this period for prehabilitation. In line with previous evidence, our participants suggested implementing remote exercise instruction and supervision through online platforms.[11, 31-34]” (line 430-439)

Comment: Discussion: Strengths of the Study:- The strengths outlined are compelling, particularly the mixed-methods approach and the high completion rate of surveys. It may be beneficial to briefly discuss how these strengths enhance the validity of the findings.

Reply: We emphasized this as “These strengths enabled our study to provide a comprehensive understanding of the facilitators, barriers, and potential solutions—some of which have seldom been identified in previous research” in line 452-455.

Comment: Discussion: Limitations:- It might enhance this section to suggest specific contexts or populations where future research could be conducted to validate or expand upon your findings.

Reply: We added “Multi-center studies with larger sample sizes across diverse cultural backgrounds are warranted in the future” in line 458-459.

Comment: Discussion: Limitations:- The comment about the use of the “Physical Activity Guidelines for Americans” is important. Consider providing a brief rationale for why these guidelines were chosen, despite their limitations, and suggest how future studies might develop tailored guidelines for this population.

Reply: We revised this as “However, these guidelines were not originally developed for prehabilitation among neoadjuvant therapy patients and may need to be adapted based on their cardiorespiratory tolerability and the fitness requirements of major surgeries” (line 463-465).

Comment: Conclusions:- It might be beneficial to include a call to action, urging stakeholders (e.g., healthcare providers, policymakers) to prioritize these changes as improvement in patient education regarding exercise prehabilitation.

Reply: Thank you. We highlighted this as “To address the identified barriers, key stakeholders are encouraged to prioritize initiatives such as enhancing physician education, implementing collaborative clinics, and providing remote supervision” (line 474-476)

Reviewer 2 Report

Comments and Suggestions for Authors

Dear Editor of the Journal, Hello. Thank you for the opportunity to review your journal.

The abstract is long, please rewrite it.

The introduction is very short, the statement of the problem and the necessity of the work should be specified.

What was the sampling method based on? Anthropometric indicators should be mentioned. The questionnaires should be fully described.

The significance of the graphs should be indicated.

The discussion should be complete. The discussion should focus more on the reasons and mechanisms.

The references should be updated.

Author Response

Dear Reviewer 2,

Please find the revised version of our manuscript (No.: 3463205) entitled “Patient Education on Exercise Prehabilitation among Patients Receiving Neoadjuvant Therapy for Cancer Surgery in China: a Mixed-Methods Study”. Thank you for offering us the opportunity to revise our manuscript for further consideration by the Healthcare.

We have revised the manuscript in response to the editors’ and reviewers’ comments. Our detailed responses to each of the comments are shown below. We would like to thank the editors and reviewers for their insightful comments, which have helped us improve the manuscript. We hope that this revised version addresses all comments adequately and that our manuscript is now considered acceptable for publication in Healthcare.

Sincerely,

Xuerong Yu, MD

-----------------------

Point-by-point Response to Comments of Reviewers

Reviewer 2

Comment: The abstract is long, please rewrite it.

Reply: Thank you for your reminder. We have shortened the Methods section (lines 23-25) in the abstract, and the word count is now 246, which is below the upper limit of 250.

Comment: The introduction is very short, the statement of the problem and the necessity of the work should be specified.

Reply: We totally agree with you. We included introduction of exercise prehabilitation as “The potential efficacy of exercise prehabilitation includes improving functional capacity, optimizing organ perfusion, and promoting anti-tumor defense.[8]” (line 50-52) and introduction of neoadjuvant therapy as “In recent years, neoadjuvant therapy has been increasingly utilized in the treatment of breast, esophageal, gastric, pancreatic, rectal, extremity sarcoma, bladder, and ovarian cancers.[14]” (line 59-61). We also emphasized the necessity of the work as “Understanding these factors can help us identify areas for improvement and develop targeted strategies to enhance exercise prehabilitation” (line 72-74).

Comment: What was the sampling method based on? Anthropometric indicators should be mentioned. The questionnaires should be fully described.

Reply: We did not apply any sampling method for the survey. Instead, we consecutively included all adult patients who received neoadjuvant therapy and were scheduled for cancer surgery between May 22, 2024, and September 15, 2024. We further clarified this in line 102-104. The BMI of included patients were demonstrated in Table 1. For the qualitative part, we used a nonprobabilistic, maximum variation sampling strategy to recruit interview participants, ensuring representation from all parties involved in neoadjuvant therapy. The participants included patients and their family, surgeons from different cancer subspecialties, oncologists from both our hospital and community hospitals, anesthesiologists, rehabilitation physicians, physical therapists, and nurses. We explained this in line 179-189.

Comment: The significance of the graphs should be indicated.

Reply: We added the comparison of different sources of knowledge and different barriers in methods as “Additionally, the sources of their knowledge and potential barriers to engaging in excise prehabilitation were analyzed by the Chi-Square test.” (line 159-160), and added the p value in Figure 2B and C.

Comment: The discussion should be complete. The discussion should focus more on the reasons and mechanisms.

Reply: We provided more details on the reasons why patients did not receive education regarding exercise from physicians in line 410-420, and also provided targeted solutions in line 421-426. This paragraph was as follows: “Although rehabilitation physicians and therapists were the most qualified to offer specific guidance on exercise prehabilitation, patients often did not realize they needed to visit the rehabilitation department unless recommended or referred by their surgeons, oncologists, or anesthesiologists. However, a lack of knowledge, time, and energy presented significant challenges to these physicians. Our results showed that surgeons were the most trusted physicians by patients during neoadjuvant therapy, and their educational efforts were the most effective. Unfortunately, however, many of the surgeons in our study missed valuable opportunities to emphasize the importance of exercise prehabilitation, possibly due to their own lack of understanding of its benefits. This underscores the need for improved physician education. Another reason was that surgeons, oncologists, and anesthesiologists had limited time for each patient in their outpatient clinics, which prevented them from providing education on exercise. A recent randomized controlled trial demonstrated the effectiveness of group-based patient education, presenting another potential solution to addressing the limitation of physicians' time and energy for individualized patient.[29]”.

Comment: The references should be updated.

Reply: Thank you. We added several references that published in 2023 and 2024 (16-19, 23, and 30).

Reviewer 3 Report

Comments and Suggestions for Authors

- It would be beneficial to emphasize the distinctiveness of this study and the limitations of previous research more clearly in the background section.

- There is a question regarding whether the Physical Activity Guidelines for Americans are appropriate for patients undergoing neoadjuvant therapy. Is the recommended intensity feasible for these patients?

- A discussion on more objective methods to measure exercise performance, such as the use of wearable devices, should be included. If this is not feasible, it should be acknowledged as a study limitation.

- A more detailed description of the measures taken to minimize researcher bias during the data analysis process is necessary.

- It is important to discuss which patient groups would benefit from remote exercise supervision and which groups might face limitations.

Author Response

Dear Reviewer 3,

Please find the revised version of our manuscript (No.: 3463205) entitled “Patient Education on Exercise Prehabilitation among Patients Receiving Neoadjuvant Therapy for Cancer Surgery in China: a Mixed-Methods Study”. Thank you for offering us the opportunity to revise our manuscript for further consideration by the Healthcare.

We have revised the manuscript in response to the editors’ and reviewers’ comments. Our detailed responses to each of the comments are shown below. We would like to thank the editors and reviewers for their insightful comments, which have helped us improve the manuscript. We hope that this revised version addresses all comments adequately and that our manuscript is now considered acceptable for publication in Healthcare.

Sincerely,

Xuerong Yu, MD

-----------------------

Point-by-point Response to Comments of Reviewers

Reviewer 3

Comment: - It would be beneficial to emphasize the distinctiveness of this study and the limitations of previous research more clearly in the background section.

Reply: Thank you for your suggestion. We summarized the previous similar studies and emphasized the distinctiveness of our study in introduction as “Previous studies have illustrated facilitators and barriers of exercise prehabilitation among elderly or frail patients.[16-19] However, evidence is limited among patients undergoing neoadjuvant therapy. How effectively patients perform exercise prehabilitation during or after neoadjuvant therapy remains uncertain” (line 62-66).

Comment: - There is a question regarding whether the Physical Activity Guidelines for Americans are appropriate for patients undergoing neoadjuvant therapy. Is the recommended intensity feasible for these patients?

Reply: We agree with you that this should be a question. To our knowledge, there are no established criteria for the amount of exercise adapted to patients undergoing neoadjuvant therapy; therefore, we used the Physical Activity Guidelines for Americans instead. We acknowledged this in limitations as “However, these guidelines were not originally developed for prehabilitation among neoadjuvant therapy patients and may need to be adapted based on their cardiorespiratory tolerability and the fitness requirements of major surgeries” (line 463-465 ).

Comment: - A discussion on more objective methods to measure exercise performance, such as the use of wearable devices, should be included. If this is not feasible, it should be acknowledged as a study limitation.

Reply: Thanks. We agree that objective measurement of exercise performance was important, but it is not feasible in our study; therefore, we acknowledge this in limitations as “Third, the exercise performance data were collected through survey, which might introduce information bias. Objective measurement using wearable devices is recommended for future studies.” (line 466-469).

Comment: - A more detailed description of the measures taken to minimize researcher bias during the data analysis process is necessary.

Reply: Thank you for your constructive suggestions. We provided more details regarding reflexivity memo and peer debriefing as “Notably, X.X. and Y.X., both anesthesiologists, had professional and personal affiliations with some participants (insiders). To mitigate potential insider bias, reflexivity memo writing was employed, allowing researchers to continuously document their thoughts, biases, and positionality throughout the research process. This practice helped maintain self-awareness and ensured that interpretations were data-driven rather than influenced by prior relationships or assumptions. Additionally, peer debriefing with Y.T. and Z.Y., who were not anesthesiologists (outsiders), provided an external perspective, challenging assumptions and reducing the risk of misinterpretation due to pre-existing researcher biases” (line 228-236) .

Comment: - It is important to discuss which patient groups would benefit from remote exercise supervision and which groups might face limitations.

Reply: Thank you for raising this important point. Our data showed patients with prior exercise habit were more likely to meet the standards of exercise amount. Compared with them, patients without exercise habits may face limitations from remote exercise supervision. We explained this as “Additional attention and protection should be given to patients without prior exercise habits, as they may face greater challenges in implementing exercise plans, particularly muscle-strengthening activities. They also face a higher risk of injury if they struggle to understand remote exercise supervision.” (line 442-445).

Round 2

Reviewer 2 Report

Comments and Suggestions for Authors

edited and its can be publish

Reviewer 3 Report

Comments and Suggestions for Authors

The authors have adequately addressed the reviewers' comments through appropriate revisions and responses. They have acknowledged the limitations of the study and added additional points to address them. Other responses are logical, and I respect the authors' perspectives. Therefore, I recommend accepting the manuscript in its current form.